# Incremental Doses of Nitrate-Rich Beetroot Juice Do Not Modify Cognitive Function and Cerebral Blood Flow in Overweight and Obese Older Adults: A 13-Week Pilot Randomised Clinical Trial

**DOI:** 10.3390/nu14051052

**Published:** 2022-03-02

**Authors:** Abrar M. Babateen, Oliver M. Shannon, Gerard M. O’Brien, Edward Okello, Ellen Smith, Dilara Olgacer, Christina Koehl, William Fostier, Emma Wightman, David Kennedy, John C. Mathers, Mario Siervo

**Affiliations:** 1Human Nutrition Research Centre, Population Health Sciences Institute, Newcastle University, Newcastle upon Tyne NE2 4HH, UK; a.m.o.babateen2@ncl.ac.uk (A.M.B.); oliver.shannon@ncl.ac.uk (O.M.S.); edward.okello@newcastle.ac.uk (E.O.); olgacerdilara@gmail.com (D.O.); chrissi.koechl18@gmail.com (C.K.); w.fostier@ncl.ac.uk (W.F.); john.mathers@ncl.ac.uk (J.C.M.); 2Clinical Nutrition Department, College of Applied Medical Sciences, Umm Al-Qura University, Makkah 21955, Saudi Arabia; 3Translational and Clinical Research Institute, Newcastle University, Newcastle upon Tyne NE2 4HH, UK; 4Brain Performance and Nutrition Research Centre, Northumbria University, Newcastle upon Tyne NE1 8ST, UK; ellen.smith@northumbria.ac.uk (E.S.); emma.l.wightman@northumbria.ac.uk (E.W.); david.kennedy@northumbria.ac.uk (D.K.); 5Nutrition Trials at Northumbria (NUTRAN), Northumbria University, Newcastle upon Tyne NE1 8ST, UK; 6School of Life Sciences, Queen’s Medical Centre, The University of Nottingham Medical School, Nottingham NG7 2UH, UK

**Keywords:** inorganic nitrate, beetroot juice, cognition, cerebral blood flow, older adults

## Abstract

Nitrate-rich food increases nitric oxide (NO) production and may have beneficial effects on vascular, metabolic, and brain function. This pilot study tested the effects of prolonged consumption of a range of doses of dietary nitrate (NO_3_^−^), provided as beetroot juice, on cognitive function and cerebral blood flow (CBF) in overweight and obese older participants. The study had a 13-week single-blind, randomised, parallel design, and 62 overweight and obese older participants (aged 60 to 75 years) received the following interventions: (1) high NO_3_^−^ (2 × 70 mL beetroot juice/day) (2) medium NO_3_^−^ (70 mL beetroot juice/day), (3) low NO_3_^−^ (70 mL beetroot juice on alternate days), or (4) placebo (70 mL of NO_3_^−^-depleted beetroot juice on alternate days). Cognitive functions were assessed using the Computerised Mental Performance Assessment System (COMPASS) assessment battery. CBF, monitored by concentration changes in oxygenated and deoxygenated haemoglobin, was assessed in the frontal cortex using near-infrared spectroscopy. The findings of this pilot study showed that cognitive function and CBF were not affected by supplementation with NO_3_^−^-rich beetroot juice for 13 weeks, irrespective of the NO_3_^−^ dose administered. These findings require confirmation in larger studies using more sophisticated imaging methods (i.e., MRI) to determine whether prolonged dietary NO_3_^−^ supplementation influences brain function in older overweight people.

## 1. Introduction

Ageing is associated with a progressive decline in cognitive, metabolic, and cardiovascular functions [1,2,3]. Vascular risk factors, including hypertension and obesity, are independently related to increased risk of cognitive decline and dementia in middle-aged and older adults [4]. The association between cognitive impairment and reduced cerebral blood flow (CBF) has been reported in several studies [5,6,7,8], and reduced CBF may contribute to the development of dementia [8,9]. Data from 1730 older individuals (aged 55 years or older) showed that higher baseline CBF was associated with a lower risk of cognitive decline and dementia diagnosis after a 6.5-year follow-up [10]. In addition, a recent observational study showed that higher CBF was associated with improved executive, attention, and memory function in 452 older white Europeans [11].

Nitric oxide (NO) is a gaseous and highly reactive molecule that can diffuse easily to surrounding tissues. NO is produced via the activity of inter-connected enzymatic and non-enzymatic pathways [12]. The former pathway utilises arginine as a precursor, which is converted into equimolar amounts of NO and citrulline by NO synthase, existing in three different function-specific isoforms (i.e., endothelial, inducible, and neuronal). Optimal functioning of this pathway requires various cofactors, including NADPH, calmodulin, and tetrahydrobiopterin [12]. The non-enzymatic pathway includes the action of oral commensal bacteria that reduce NO_3_^−^ into NO_2_^−^, which is then converted into NO under conditions of low pH (i.e., in the stomach) or low oxygen tension (i.e., in peripheral microcirculation) [13]. NO is a neurotransmitter regulating multiple neurobiological processes, such as the regulation of synaptic plasticity [12]. NO production decreases with ageing, which is likely to contribute to degenerative processes of the cardiovascular and central nervous systems [14]. Ageing-related NO insufficiency is multifactorial, but oxidative stress has been suggested as the primary factor [15,16]. NO is scavenged by reactive oxygen species, mainly superoxide, to form a potent oxidant called peroxynitrite [17] that reduces NO concentration in cells [18]. Peroxynitrite generation is one of the crucial pathogenic mechanisms in neurodegenerative disorders [19]. 

Nutrition plays a key role in the development, maintenance, and treatment of mental health and related disorders, whose importance has been unified under the umbrella term nutritional psychiatry [20,21]. Nutritional and lifestyle interventions capable of maintaining or restoring normal NO production may reduce the risk of atherosclerosis and cardiovascular diseases [22]. In turn, this may reduce the risk of cognitive impairment and dementia, since better cardiovascular health is associated with a lower rate of cognitive decline and risk of dementia [23]. For instance, polyphenols and polyunsaturated fatty acids may improve brain health via their anti-inflammatory properties and increased NO availability [24]. Specifically, the two NO pathways can be targeted by nutritional and dietary interventions that boost NO production via the enzymatic (i.e., arginine, citrulline, polyphenols, and tetrahydrobiopterin cofactor) or non-enzymatic (NO_3_^−^ or NO_2_ supplementation or NO_3_^−^-rich foods, such as beetroot, cabbage, spinach, or rocket) pathways [13]. 

NO_3_^−^ supplementation may provide a non-pharmacological dietary strategy with which to improve CBF. Recent studies have reported improvements in cognitive function (executive performance) and motor skills after dietary NO_3_^−^ supplementation, which appear to be mediated by augmented CBF and the efficiency of neuronal metabolism [25,26]. However, Clifford et al. [27] conducted a systematic review and meta-analysis on the topic, which indicated a lack of convincing evidence for the effects of nitrate (NO_3_^−^) supplementation on cognitive function and CBF. Overall, trials were characterized by small sample sizes (<30 participants), short duration (only one study had a duration of 10 weeks [28], whereas the remaining trials had a duration of less than two weeks), and most of the trials included healthy, normal-weight participants. In contrast, a recent study found that the consumption of raw beetroot for eight weeks improved components of cognitive function, including sustained attention, processing speed, and fatigue resistance, in older patients with diabetes [29]. The presence of effects in this sample could be due to the poorer health of these participants, including being overweight. It is now well established that obesity is associated with endothelial dysfunction, which is characterised by NO insufficiency [30,31].

We hypothesised that prolonged supplementation with dietary NO_3_^−^ would enhance brain health in older obese and overweight individuals, who are at greater risk of cognitive dysfunction. This pilot RCT was designed primarily to determine the feasibility and acceptability of the protocol for a 13-week intervention study in which overweight and obese older participants were asked to consume a range of doses of NO_3_^−^-rich beetroot juice for 13 weeks, and the primary outcomes for this have been described elsewhere [32]. The secondary aim of the study tested whether the different doses of dietary NO_3_^−^ result in changes in cognitive, vascular, and pulmonary functions and CBF. This paper focuses on cognitive function and CBF.

## 2. Materials and Methods

### 2.1. Ethical Approval and Consent

The study (ISRCTN14746723) was approved by the Faculty of Medical Sciences, Newcastle University (1504/4477/2018). Potential participants were asked to read and sign the consent form before entering the study.

### 2.2. Participants and Study Design

This was a 13-week, randomized, single-blind, placebo-controlled, four-arm parallel feasibility trial. Male and female older participants (60 to 75 years old) with a body mass index (BMI) range between 25 and 40 kg/m^2^ were included in the study. A detailed list of the inclusion and exclusion criteria is reported elsewhere [32]. Eligible participants were randomized into one of four treatments:High NO_3_^−^: two 70 mL shots of concentrated beetroot juice per day (approximately ~400 mg of NO_3_^−^ per shot, as reported by manufacturer (James White Company, Ashbocking, Suffolk, UK)), one every morning (~8 am) and one every evening (~9 pm).Medium NO_3_^−^: one shot of concentrated beetroot juice every evening (~9 pm).Low NO_3_^−^: one shot of concentrated beetroot juice every other evening (~9 pm).Placebo: one shot of NO_3_^−^-depleted beetroot juice (~0.001 mg of NO_3_^−^) every other evening (~9 pm).

All included participants were cognitively healthy with no self-reported history of cognitive impairment. 

### 2.3. Study Setting

The study visits were conducted at the NU-Food research facility at Newcastle University. The CBF measurements were performed at the Brain Performance Nutrition Research Centre at Northumbria University.

### 2.4. Data Collection Procedures

Measurements were performed at the Newcastle University research facilities between 09:00 and 10:00. Body composition, waist circumference, and biological samples, including blood, saliva, and urine samples, as well as salivary strips, were collected. Biological sample collection procedures and related results have been reported elsewhere [32,33]. Self-reported physical activity was assessed using the short version of the international physical activity questionnaire (IPAQ). In addition, measurements of cognitive, vascular, and pulmonary functions were performed. The measurement protocols have been described in detail previously [32]. Before they left the research facility, participants were provided with a small snack (orange juice and a muffin) and were asked to continue following the low-NO_3_^−^ diet for the rest of the day. All measurements were taken in fasting condition (~12 h). Measurements of CBF performed at Northumbria University occurred between 15:00 and 16:00. Results related to peripheral vascular and pulmonary functions will be published in due course.

The end-of-study visits were performed after 13 weeks following the same order as for the baseline measurements. These two visits were scheduled to occur ~12 h after the consumption of the last beetroot juice dose for high-NO_3_^−^ and medium-NO_3_^−^ groups, and ~36 h after the consumption of the last beetroot juice dose for the low-NO_3_^−^ and placebo groups. 

Participants were asked to follow a low-NO_3_^−^ diet during the day before these visits and to limit their alcohol and caffeine consumption for 24 h before the visit. Participants were also instructed to maintain their habitual dietary habits and physical activity and to avoid using mouthwash during the study [34]. Self-reported dietary NO_3_^−^ intake for the participants (excluding the intervention supplement) during the study was also assessed via an online validated platform (Intake 24) [33].

### 2.5. Compliance with Intervention

Compliance with the intervention was assessed subjectively by completing a daily compliance log, which was used to record the time when participants consumed the beetroot juice. Compliance was also assessed objectively by measuring NO_3_^−^ and nitrite (NO_2_^−^) concentrations in plasma, saliva, and urine samples [33].

### 2.6. Measurements

The measurements described below were applied at baseline and after 13 weeks.

#### 2.6.1. Anthropometry and Body Composition

Height to the nearest 0.5 cm was measured using a stadiometer with an adjustable headpiece. Body weight and body composition parameters (fat mass, fat-free mass, body fat %, and total body water) were assessed using bioelectrical impedance analysis (Tanita BC420 MA, Tanita Corporation, Tokyo, Japan). Weight and height were used to calculate BMI. Waist circumference was measured at the midpoint between the lowest margin of the last rib and the top of the iliac crest.

#### 2.6.2. Cognitive Function

Participants were informed about the cognitive tasks and given three full practice runs at the screening visit to reduce learning effects and test anxiety prior to baseline testing day. This approach has been described by Bell et al. [35] and Goldberg et al. [36] as a solution to confounder practice effects in cognitive testing. The Computerised Mental Performance Assessment System software (COMPASS, University of Northumbria, Newcastle-upon-Tyne, www.cognitivetesting.co.uk (accessed on 31 December 2019)), which is sensitive to a range of nutritional interventions [37,38], was used to assess cognitive function. In addition, Trail Making Tasks A and B (TMT-A & TMT-B), which reportedly improved after 10 weeks of NO_2_^−^ supplementation in older adults [28], were also included. The order of the tests included in the COMPASS software were: word presentation, immediate word recall, numeric working memory, choice reaction time, Stroop, digit vigilance, computerized corsi blocks, peg and ball, delayed word recall, and word recognition. The scores from appropriate tasks were combined to deliver measures of the following cognitive domains, namely, (a) accuracy of attention, (b) speed of attention, (c) working memory, (d) episodic memory, (e) speed of memory, (f) overall accuracy, and (g) overall speed, using the method described recently by Wightman et al. [39]. Figure 1 shows how these global cognitive measures were compiled from individual task sub-scores. The tasks were presented on a laptop PC with responses made either with the keyboard or using a mouse. A description of these cognitive tasks is provided in Appendix A.

Figure 1 shows the order in which each of the cognitive tasks was completed at baseline and after 13 weeks with the approximate task timings and predominant cognitive domain of each task. It also demonstrates how measures of each of the global cognitive domains were constructed.

#### 2.6.3. Quantitative Near-Infrared Spectroscopy (qNIRS)

A frequency domain “quantitative” NIRS system (OxiplexTS Frequency-Domain Near-Infrared Tissue Oximeter; ISS, Inc., Champaign, IL, USA) was used to measure CBF. NIRS has been used widely in neuroscience research to detect changes in cerebral blood oxygenation related to brain activity [40]. It allows for the quantification of oxygenated haemoglobin (HbO_2_) and deoxygenated haemoglobin (HHb) by giving the absolute measurements of absorption of near-infrared light emitted at two different wavelengths. From these values, total haemoglobin (THb) and oxygen saturation (Ox%) can be determined as follows: THb = HbO_2_ + HHB and Ox% = HbO_2_/THb.

After attaching the sensors to the participant’s forehead, a 5 min resting measurement of CBF parameters was recorded. Then, participants performed cognitive tasks known to activate the prefrontal cortex of the brain (serial subtraction 3 (3:20 min) and 7 (3:20 min), Stroop (3:00 min), and peg and ball (2:30 min) tasks), which are associated with an activation of the prefrontal cortex [41,42,43]. The qNIRS assessment timeline is described in Figure 2.

The qNIRS device was located in the Brain Performance Nutrition Centre, Northumbria University, and, therefore, this measurement was conducted on a separate day from the main cognitive function measurements that were conducted at Newcastle University. The measurement was performed between 15.00 and 16.00 after participants consumed a low-NO_3_^−^ meal around 12.00. 

### 2.7. Sample Size Calculation

This was a pilot study designed to assess the feasibility and acceptability of the proposed intervention. A sample size of 15 per intervention group was based on (i) the predicted effect size of the intervention on cognitive changes (Trail Making Task B) [28] and (ii) the guidelines indicated by Whitehead et al. [44], who provided guidance on sample size calculation for pilot studies with the aim of maximising the use of resources and avoiding type II errors. Specifically, a sample size of 15 individuals per group would provide a 90% power to detect a medium effect size between 0.3 and 0.7.

### 2.8. Statistical Methods

The Statistical Package for Social Sciences (IBM SPSS, version 23, Armonk, NY, USA) was used to perform the analysis. Data were checked for normality by visual inspections of histograms and by the Shapiro–Wilk test. One-way ANOVA tests were used to assess differences between the four intervention groups at baseline. Participants who did not complete the study were excluded from the main analyses; the baseline characteristics of non-completers were compared to participants with complete data using an independent t-test to identify the risk of selection bias. 

To assess the effect of the 13-week intervention on cognitive performance, the 13-week data were used to calculate change from baseline for each cognitive score. A one-way ANOVA was then applied to evaluate differences between the intervention groups for each calculated change in cognitive score. The Dunnett test was conducted to compare active NO_3_^−^ groups to the placebo control group. Summary data are presented as means ± SEM.

Regarding NIRS data analysis, the data were averaged across the two hemispheres. To test the effect on the resting CBF, data recorded at resting time (5 min) at 13 weeks were averaged and converted to establish the change from resting period at baseline, and a one-way ANOVA was then used to analyse the data. The NIRS data recorded during performing cognitive tasks were converted to the change from the resting period (average of the 5 min) at baseline, and then averaged into 4 different epochs for analysis. Epoch 1, 2, 3, and 4 represent the average of the data collected when performing serial subtraction 3 (3:20 min), serial subtraction 7 (3:20 min), Stroop (3 min), and peg and ball (2:30 min), respectively. Then, analysis was conducted via a repeated measures ANOVA, utilising the interventions and epochs as factors. Models were tested for sphericity using Mauchly’s test, and multivariate models were applied if these assumptions were violated. The Dunnett test was conducted to compare active NO_3_^−^ groups to placebo control group. Statistical significance was set at *p* < 0.05. Summary data are presented as means ± SEM.

## 3. Results

### 3.1. Baseline Characteristics

Baseline characteristics of the trial participants are shown in Table 1. There were 24 (38.7%) males and 38 (61.3%) females, with a mean age 66 years (range: 60–73). BMI ranged from 25 to 39 kg/m^2^, with a mean of 30.4 kg/m^2^. Thirty-five participants (56%) were overweight and twenty-seven (44%) were obese. Intervention groups were well matched across all variables. Although sixty-two participants were enrolled in the study, twelve participants dropped out. The baseline characteristics of the 12 participants who dropped out did not differ from those of the participants who completed the study (data are shown in Appendix A). 

### 3.2. Effects of Nitrate Supplementation on Measures of Cognitive Function

The examination of the change from baseline (using an ANOVA) did not reveal any significant differences between intervention groups for any of the individual measures of cognitive function. In addition, when scores for the individual tests were combined to provide estimates of global cognitive measures, there were no significant treatment effects. The effects of NO_3_^−^ dose on each cognitive measure and on the global cognitive domains are summarized in Table 2 and Table 3, respectively.

### 3.3. Effects of Nitrate Supplementation on Measures of qNIRS Parameters Used to Estimate Cerebral Blood Flow

No significant difference was found between the two resting periods at baseline and after 13 weeks for any CBF-related parameter (*p* > 0.05). The repeated measures ANOVA did not reveal any significant difference with epochs (cognitive tasks used; serial sub3, serial sub7, Stroop, and peg and ball) (*p* > 0.05), intervention group (*p* > 0.05), or the interaction between epoch and intervention group (*p* > 0.05) on any of the CBF parameters. Since no significant difference was found between epochs, data for all epochs within each participant were combined. Figure 3A–D show the effect on CBF-related measures including oxygen saturation, total haemoglobin, oxyhaemoglobin, and deoxyhaemoglobin. The mean values of qNIRS parameters for each intervention group at baseline and after 13 weeks are reported in Appendix A.

### 3.4. Plasma Nitrate and Nitrite Concentrations

Other results have been reported elsewhere [33]. Briefly, a one-way ANOVA revealed that there was a statistically significant increase in the change from baseline in plasma NO_3_^−^ concentration at 13 weeks (*p* < 0.001) with increasing beetroot juice consumption. However, evidence for a change from baseline in plasma NO_2_^−^ concentration was less convincing (*p* = 0.054). Data are summarised in Appendix A.

## 4. Discussion

### 4.1. Summary of Main Findings

To date, this is one of the largest and longest-duration RCTs investigating the effects of incremental doses of supplemental NO_3_^−^ in the form of beetroot juice on cognitive function and CBF in overweight and obese older adults. This study was designed primarily as a feasibility study and was not intended to provide a definitive investigation of the effects of supplemental NO_3_^−^ on CBF and cognitive function. With that proviso, there was no evidence that cognitive function or CBF measured by NIRS were influenced by NO_3_^−^ supplementation at any dose.

### 4.2. Effects of Prolonged Beetroot Juice Consumption on Cognition and on CBF

Beetroot juice is a rich source of NO_3_^−^, which is converted to NO following consumption. NO plays an important role in the regulation of CBF, neurotransmission, and neurovascular coupling [45,46]. Ageing is associated with endothelial dysfunction, which is characterised by a lowered release of endothelial NO. As a consequence, cerebral haemodynamics can be disturbed by reduced NO availability and higher reactive oxygen species production [47]. Lowered CBF is widely acknowledged as a key contributor to cognitive decline [48,49]. The present study is the first to investigate the effect of 13 weeks of consumption of beetroot juice as a rich source of NO_3_^−^ on cognition and CBF in older overweight and obese individuals who are at greater risk of endothelial dysfunction and cognitive impairment. 

The study found no effect of supplementation with beetroot juice on resting CBF or stimulated CBF (during the performance of cognitive tasks that activate the prefrontal cortex) following high-NO_3_^−^, medium-NO_3_^−^, or low-NO_3_^−^ doses in comparison with a placebo. It has previously been reported that acute dietary NO_3_^−^ supplementation with beetroot juice increased CBF in young healthy adults, when measured using a similar approach (NIRS) to the one used in the current study [25]. However, evidence regarding possible associations between dietary NO_3_^−^ and CBF in older people is very limited. Working with older adults, Presley et al. found that resting regional white matter perfusion (dorsolateral prefrontal cortex only), assessed via magnetic resonance imaging, was improved following two days of high-NO_3_^−^ diet supplementation [26]. In contrast, Kelly et al. did not detect any changes in apparent diffusion coefficients in the aforementioned regions of the brain following three days of NO_3_^−^ supplementation in older adults [50], despite the fact that a larger dose of NO_3_^−^ was used than that employed in the Presley et al. study. With regard to this apparent disparity in outcomes, Kelly et al. have postulated that the participants in the Presley et al. study, who were on average 10 years older than those in the Kelly et al. study, may have been more responsive to dietary NO_3_^−^. The same rationale might explain the lack of observed effect in the present study, where the participants were—on average—eight years younger than those in the Presley et al. study. In normotensive and hypertensive subjects, the forearm vasodilatory response to acetylcholine measured by strain-gauge venous plethysmography was inversely associated with ageing [51].

Alternatively, the lack of effect on CBF observed in the present study may be explained by a potential reduction in NO formation with longer term NO_3_^−^ supplementation. As noted, NO is synthesised via enzymatic (L-arginine–NOS pathway) and non-enzymatic (NO_3_^−^–NO_2_^−^–NO) pathways [13]. Long-term dietary NO_3_^−^ supplementation in rats results in the reversible down-regulation of endothelial nitric oxide synthase activity, suggesting cross-talk between the two pathways [52]. We hypothesise that, after continuous NO_3_^−^ intake for three months, as in the present study, the endogenous pathway was down-regulated. This hypothesis could be tested by quantifying whole-body NO production before and after long-term NO_3_^−^ supplementation.

Findings from the present study provided no evidence that the 13-week ingestion of different doses of NO_3_^−^ in the form of beetroot juice had any beneficial effect on cognitive function in our participants. The only statistically significant difference found was restricted to a single task outcome (delayed word recall), which most likely represents a chance finding. Gilchrist and co-authors were the first to report the beneficial effects of beetroot juice ingestion on cognitive performance, specifically in simple reaction time, in older adults with type-II diabetes following 14 days of beetroot juice consumption [53]. Justice et al. reported a 14% and 18% improvement in TMT-B following 10 weeks of consumption of 80 mg/d and 160 mg/d NO_2_^−^ in older adults, respectively [28]. Although the present study was 11 weeks longer than the Gilchrist et al. study and three weeks longer than the Justice et al. study, no such beneficial effects were found with any of the cognitive outcomes. The lack of cognitive effects in the current study may be attributable to several factors, including a lack of effect on CBF, and is consistent with the findings from our earlier systematic review [27]. Nevertheless, the inclusion of interim assessments in any future chronic supplementation study would pick up on any earlier effects, if they exist.

A recent study found a positive association between CBF and executive function in a multi-ethnic older cohort [11]. However, improvement in CBF is not always associated with improvement in cognitive performance [38]. Because CBF was not measured, it is not known whether improved CBF mediated the improvements in cognitive function following NO_3_^−^ supplementation in the Gilchrist et al. (2014) and Justice et al. (2015) studies. Furthermore, it is possible that health status influences the impact of NO_3_^−^ supplementation on cognitive performance. The diabetic participants included in the Gilchrist et al. (2014) study had deficits in cognitive function compared with healthy participants, which may have amplified their sensitivity to cognitive testing. Although the participants in the present study were apparently healthy, they had a raised BMI, which is associated with impaired NO availability [31]. In addition, raised BMI is negatively correlated with global cognitive performance in long-term prospective cohort studies [54,55]. 

### 4.3. Strengths and Limitations of the Study

Few studies have tested the effect of dietary NO_3_^−^ in older overweight/obese people, and most studies to date regarding that age group have featured short-duration intervention periods. The strength of the present study was the prolonged duration of the intervention; this is one of the longest RCTs that has investigated the effect of NO_3_^−^ supplementation on CBF and cognitive function in older overweight and obese adults. Thus, this study contributes to the small but growing body of research on the effects of dietary NO_3_^−^ supplementation on brain function. Further, the use of incremental doses of NO_3_^−^ in the form of beetroot juice during such a period, with the low dose being consumed on alternate days during the study, gave the study a unique design feature. Our participants were asked to maintain their habitual dietary habits, physical activity, and medication throughout the study. Finally, the increase in concentrations of NO_3_^−^ and NO_2_^−^ biomarkers with increasing beetroot juice dose (as reported elsewhere [33]) provide assurance that the participants were compliant in consuming the beetroot juice supplements.

On the other hand, this study had limitations. The CBF assessment was conducted on a separate study day from other measurements due to the different location of the NIRS apparatus. However, participants were asked to follow similar instructions to the previous study day that included following a low-NO_3_^−^ diet and avoiding alcohol and caffeine consumption for 24 h before the visit. The only significant difference was that participants were not required to fast before the CBF assessment. Participants were asked to have their low-NO_3_^−^ lunch ~4 h before the assessment. In addition, due to limited access to the NIRS equipment (we had only less than one hour to collect the data), only a short period was available to collect NIRS data compared with other studies that have used similar equipment [25,39,56], which might be not sufficient to see CBF parameters alterations. Furthermore, during the data collection period, NIRS equipment stopped working for 2 weeks, which led to the loss of follow up data for three participants. The most important was that this study was designed primarily to test the feasibility and acceptability of the intervention in older adults and was not designed (or powered) to make definitive conclusions regarding the prolonged effect of NO_3_^−^ on cognition and CBF. 

## 5. Conclusions

These preliminary findings indicate no beneficial effects of NO_3_^−^ supplementation on cognitive performance or CBF in older overweight and obese adults. The findings from this study will help to inform the design of future larger definitive trials to investigate the long-term effects of dietary NO_3_^−^ supplementation on cognitive functions and CBF. Although this study was conducted on an apparently healthy at-risk population (i.e., older overweight and obese individuals), studies in those with mild cognitive impairment or hypertension may be more sensitive to the beneficial effects of prolonged beetroot juice consumption on cognitive function.

## Figures and Tables

**Figure 1 nutrients-14-01052-f001:**
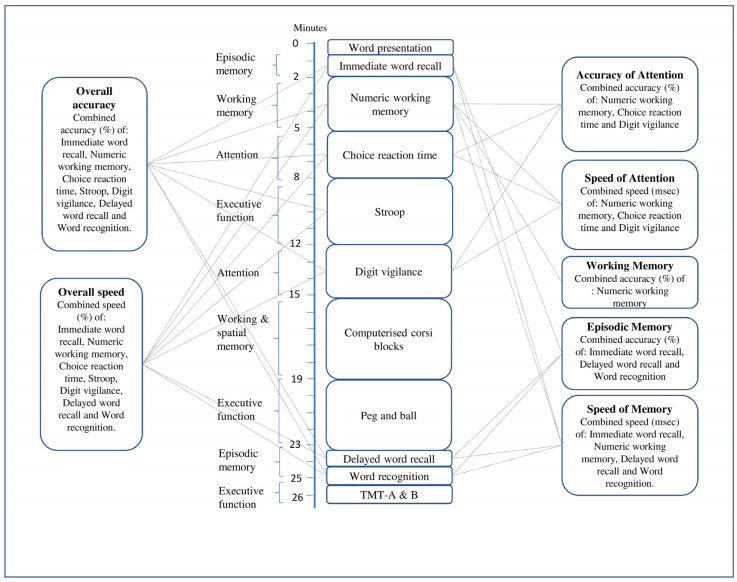
Cognitive task battery.

**Figure 2 nutrients-14-01052-f002:**
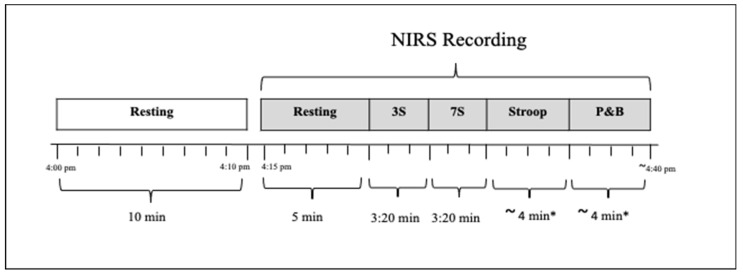
Protocol of the near-infrared spectroscopy (NIRS) assessment; 3S: serial subtraction 3, 7S: serial subtraction 7, P&B: peg and ball. * Stroop and peg and ball are not based on a fixed time, so the data were extracted based on the least time recorded by a participant.

**Figure 3 nutrients-14-01052-f003:**
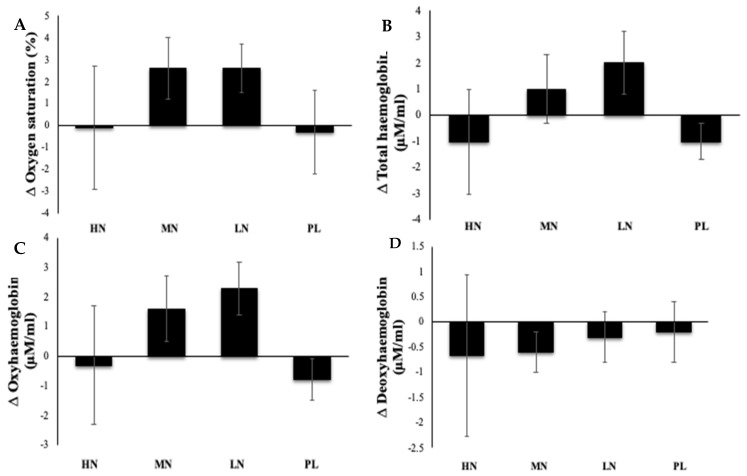
Prolonged effect of incremental doses of supplemental NO_3_^−^ in form of beetroot juice on cerebral oxygen saturation (**A**), total haemoglobin (**B**), oxyhaemoglobin (**C**), and deoxyhaemoglobin (**D**). HN, high nitrate; MN, medium nitrate; LN, low nitrate; PL, placebo. No significant differences between groups were found for any of the cerebral blood flow (CBF)-related parameters (*p* > 0.05). HN (high NO_3_^−^; two 70 mL shots of beetroot juice/day, morning and evening), MN (medium NO_3_^−^; 70 mL of beetroot juice/day), LN (low NO_3_^−^; 70 mL of beetroot juice every alternate days), and PL (placebo; 70 mL of NO_3_^−^-depleted beetroot juice every alternate days). Each shot of beetroot juice contains 400 mg of NO_3_^−^. Data represent the change of CBF parameters recorded when performing cognitive tasks at 13 weeks from the resting period at baseline and were analysed using a one-way ANOVA. Data are expressed as mean ± SEM, (*n* = 47).

**Table 1 nutrients-14-01052-t001:** Baseline characteristics of the study participants [33].

Characteristics	All	HN	MN	LN	PL
Number	62	16	17	14	15
Gender, M/F	24/38	10/6	5/12	4/10	5/10
Age (years)	66.3 ± 3.7	64.7 ± 3.5	66.7 ± 4.2	67.3 ± 2.7	65.7 ± 3.9
Education (years)	15.3 ± 3.0	16.0± 3	15.7± 2.6	15.0 ± 3.1	14.6 ± 3.1
Body weight (kg)	84.9 ± 12.6	90.9 ± 13.4	84.6 ± 10.5	80.1 ± 12.5	83.9 ± 12.6
BMI (kg/m^2^)	30.3 ± 3.7	30.5 ± 3.6	30.5 ± 3.2	29.9 ± 3.4	30.3 ± 4.8
WC (cm)	102.4 ± 9.2	104.5 ± 10.3	100.6 ± 9.6	102.1 ± 8.9	102.6 ± 8.2
FM (kg)	32.4 ± 8.7	32.1 ± 8.7	34.5 ± 9.0	31.3 ± 7.2	31.5± 10.0
FFM (%)	37.8 ± 7.8	35.2 ± 8.2	39.2 ± 7.8	39.3 ± 6.6	37.1 ± 8.1
TBW (kg)	38.6 ± 6.3	41.3 ± 7.4	38.6 ± 4.5	35.6 ± 6.3	38.8 ± 5.9
SBP (mm Hg)	135.1 ± 14.7	130.8 ± 12.0	136.1 ± 10.4	139.5 ± 13.2	134.1 ± 12.9
DBP (mm Hg)	76.9 ± 9.4	75.8 ± 9.7	77.3 ± 9.1	77.8 ± 8.1	76.9 ± 11.2
PA (METs/wk)	3667 ± 5604	2741 ± 1522	3257 ± 1845	2262 ± 1933	6280 ± 10,512

M/F, male/female; BMI, body mass index; WC, waist circumference; FM, fat mass; FFM, fat-free mass; SBP, systolic blood pressure; DBP, diastolic blood pressure; PA, physical activity. Data are expressed as mean ± SD. *p* values are based on one-way ANOVA, except for gender, which was based on Chi square test. HN, high nitrate; MN, medium nitrate; LN, low nitrate; PL, placebo.

**Table 2 nutrients-14-01052-t002:** Mean values for baseline and for change from baseline after 13 weeks of intervention on individual cognitive tasks for different doses of nitrate.

Measure	High NO_3_^−^ (*n* = 10)		Medium NO_3_^−^ (*n* = 13)		Low NO_3_^−^ (*n* = 14)		Placebo (*n* = 13)		*p* Value
	Baseline	13 Weeks (Change)	Baseline	13 Weeks (Change)	Baseline	13 Weeks (Change)	Baseline	13 Weeks (Change)	
IWR correct (number)	5.4 ± 0.4	0.05 ± 0.5	6.5 ± 0.4	0.2 ± 0.3	5.6 ± 0.5	−0.07 ± 0.4	6.8 ± 0.5	−0.7 ± 0.6	0.45
IWR error (number)	0.2 ± 0.2	0.0	0.4 ± 0.2	0.1 ± 0.2	0.6 ± 0.2	0.01 ± 0.1	0.3 ± 0.2	−0.1 ± 0.1	0.83
NWM (% accuracy)	98.0 ± 0.3	−0.23 ± 1.5	93.8 ± 1.7	1.2 ± 0.7	96.5 ± 1.4	−1.3 ± 1.6	91.1 ± 3.4	1.3 ± 1.4	0.61
NWM-RT (ms)	1126 ± 81	3 ± 37	1288 ± 179	−185 ± 143	1087 ± 77	63 ± 61	1091 ± 48	−78 ± 46	0.76
CRT (% accuracy)	98.4 ± 0.5	−0.6 ± 0.6	98.0 ± 0.5	0.8 ± 0.6	98.7 ± 0.5	−0.1 ± 0.4	98.8 ± 0.5	−0.6 ± 0.5	0.49
CRT RT (ms)	570 ± 37	−18 ± 21	622 ± 30	−31 ± 27	641 ± 46	−4 ± 0.5	603 ± 27	−6 ± 21	0.21
Stroop % accuracy *	99.3 ± 0.4	0.29 ± 0.3	98.8 ± 0.8	6.5 ± 7.1	99.9 ± 0.1	−0.2 ± 0.2	99.8 ± 0.1	−1.9 ± 10	0.79
Stroop RT (ms) *	1304 ± 28	−45 ± 48	1434 ± 70	128 ± 195	1523 ± 174	−40 ± 74	1261 ± 38	51 ± 159	0.78
DV (% accuracy)	96.5 ± 1.4	−2.5 ± 1.7	90.4 ± 2.2	−7.2 ± 7.9	86.1 ± 3.7	3.8 ± 2.3	84.8 ± 4.6	−0.89 ± 2.0	0.49
DV RT (ms)	483 ± 11	−2.8 ± 7.7	469 ± 10	−41.5 ± 38	512 ± 11	33 ± 40	498 ± 8	−0.03 ± 4	0.36
DV false alarms (number)	1.3 ± 0.3	0.7 ± 1.0	4.2 ± 0.7	−0.3 ± 1.0	5.7 ± 1.1	0.2 ± 0.8	5.6 ± 1.2	−0.4 ± 0.8	0.79
Corsi blocks span	5.6 ± 0.1	0.1 ± 0.1	5.4 ± 0.2	0.1 ± 0.2	5.2 ± 0.4	0.3 ± 0.3	5.4 ± 0.3	−0.3 ± 0.4	0.29
P&B thinking time	4106 ± 582	−367 ± 349	5221 ± 429	−1066 ± 382	4600 ± 317	−695 ± 263	5092 ± 541	−653 ± 345	0.46
P&B working time (ms)	11,640 ± 747	−974 ± 460	14,311 ± 980	−2325 ± 893	14,186 ± 1675	−1882 ± 144	12,624 ± 729	−963 ± 641	0.59
P&B errors	3.7 ± 2.5	0.8 ± 1.5	2.2 ± 1.9	−0.2 ± 0.8	4.2 ± 4.6	−1.3 ± 1.6	3.3 ± 3.3	−2.3 ± 1.1	0.48
DWR correct (number)	3.8 ± 0.5	0.3 ± 0.5	5.3 ± 0.4	−0.8 ± 0.3	3.5 ± 0.5	0.8 ± 0.3	5.0 ± 0.5	−0.1 ± 0.6	0.03
DWR error (number)	0.6 ± 0.2	0.4 ± 0.2	0.8 ± 0.2	0.4 ± 0.1	0.6 ± 0.2	0.1 ± 0.1	0.6 ± 0.2	0.1 ± 0.3	0.65
WR (%accuracy)	77.0 ± 7.9	−1.3 ± 4.3	79.4 ± 8.1	0.7 ± 2.2	78.6 ± 12.7	−0.7 ± 3.6	82.4 ± 7.8	−2.2 ± 2.7	0.77
WR RT (ms)	1277 ± 187	−14 ± 103	1085 ± 191	17 ± 39	1226 ± 316	6.1 ± 52	1287 ± 480	65 ± 48	0.39
TMT-a (s)	24.1 ± 1.7	−2.4 ± 1.5	28.1 ± 1.8	−2.5 ± 2.2	28.5 ± 2.7	−0.1 ± 1.8	24.6 ± 1.5	−1.2 ± 1.3	0.75
TMT-b (s)	52.1 ± 3.6	−8.1 ± 3.0	52.2 ± 4.2	2.2 ± 4.9	56.3 ± 5.5	−7.1 ± 2.4	55.0 ± 4.5	−9.7 ± 3.9	0.11

Data are presented as means ± SEM. Data were analysed using one-way ANOVA. *p* value is for comparison of the change in score between different study arms. * Due to non-compliance with the task instructions, data of some participants were removed. HN: high NO_3_^−^ (baseline: *n* = 13, 13 weeks: *n* = 9), MN: medium NO_3_^−^ (baseline: *n* = 15, 13 weeks: *n* = 11), LN: low NO_3_^−^ (baseline: *n* = 13, 13 weeks: *n* = 13), PL: placebo (baseline: *n* = 14, 13 weeks: *n* = 11). IWR: immediate word recall, NWM: numeric working memory, CRT: choice reaction time, DV: digit vigilance, P&B: peg and ball, DWR: delayed word recall, WR: word recognition, RT: reaction time.

**Table 3 nutrients-14-01052-t003:** Mean values for baseline and for change from baseline after 13 weeks of intervention on global cognitive measures for different doses of nitrate.

Measure	High NO_3_^−^ (*n* = 10)		Medium NO_3_^−^ (*n* = 13)		Low NO_3_^−^ (*n* = 14)		Placebo (*n* = 13)		*p* Value
	Baseline	13-Weeks (Change)	Baseline	13-Weeks (Change)	Baseline	13-Weeks (Change)	Baseline	13-Weeks (Change)	
Accuracy of attention (%)	96.6 ± 0.8	−1.8 ± 0.8	93.2 ± 1.3	−2.2 ± 2.3	94.2 ± 1.4	−0.8 ± 1.0	91.7 ± 2.2	7.2 ± 7.2	0.88
Speed of attention (msec)	735 ± 36	−6.1 ± 13.5	788 ± 54	−95.7 ± 52.3	763 ± 37	17.9 ± 21.3	742 ± 21	32 ± 62.8	0.16
Accuracy of working memory (%)	96.3 ± 1.7	−0.2 ± 1.5	91.6 ± 2.8	1.2 ± 0.7	96.4 ± 1.2	−1.3 ± 1.7	91.3 ± 3.2	8.0 ± 6.8	0.49
Accuracy of episodic memory (%)	45.2 ± 1.9	0.7 ± 3.1	52.0 ± 2.1	−1.2 ± 1.8	46.3 ± 2.5	1.4 ± 1.3	54.8 ± 2.9	2.5 ± 6.1	0.42
Speed of memory (msec)	1205 ± 51	−5.6 ± 44.3	1186 ± 72	−83.7 ± 67.2	1155 ± 66	34.7 ± 32.1	1214 ± 76	88.2 ± 97.5	0.37
Overall speed (msec)	947 ± 25	−15.8 ± 15.2	969 ± 37	−42.3 ± 40.6	997 ± 57	2.5 ± 25.1	954 ± 35	76.3 ± 73.1	0.77
Overall accuracy (%)	74.6 ± 1.2	0.7 ± 1.6	75.7 ± 1.3	−1.1 ± 1.6	73.9 ± 1.4	0.2 ± 0.6	76.9 ± 1.5	4.8 ± 6.6	0.53

Data are presented as means ± SEM. Data were analysed using one-way ANOVA. *p* value is for comparison of the change in score between different study arms.

## Data Availability

Not applicable.

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
