# Peer review of "Incremental Doses of Nitrate-Rich Beetroot Juice Do Not Modify Cognitive Function and Cerebral Blood Flow in Overweight and Obese Older Adults: A 13-Week Pilot Randomised Clinical Trial"

_nutrients, 2022, doi:10.3390/nu14051052_

Round 1

Reviewer 1 Report

It is an interesting study to that incremental doses of nitrate-rich beetroot juice did not modify cognitive function and CBF in overweight and obese older adults for 13 weeks. 
1. The clarity of the work was further hampered by the language and grammar in the writing.
2. Unfortunately, the authors have described various research papers, but fail draw meaningful discussion how this would be important for their finding.
3. The authors try to think about how to improve Fig. 1 since it presents really complicated and confused (many cross-line are overlapping...)
4. qNIRS was applied to veryfy their data. The NIRS assessment timeline need to provided. I do agree that thier data need to be additional confirmed by other more sophisticated imaging methods (i.e., MRI), thus I ask the authors provide addtional acute effects on cerebral blood flow (CBF) parameters after the first dose treament. 
5. Also, effects of the dose on hemodynamic responses to brain activation (task performance) on day 1 and day 13 of treatment, that should be provided clearly

Author Response

Reviewer 1

It is an interesting study to that incremental doses of nitrate-rich beetroot juice did not modify cognitive function and CBF in overweight and obese older adults for 13 weeks. 
1. The clarity of the work was further hampered by the language and grammar in the writing.

Thank you for this comment. The manuscript has been carefully reviewed and edited.

  1. Unfortunately, the authors have described various research papers, but fail draw meaningful discussion how this would be important for their finding.

We have carefully reviewed the paper and amended the discussion to enhance clarity.

  1. The authors try to think about how to improve Fig. 1 since it presents really complicated and confused (many cross-line are overlapping...)

Many thanks for this useful comment; we have made some changes to Figure1 to improve clarity.

  1. qNIRS was applied to verify their data. The NIRS assessment timeline need to provided. I do agree that their data need to be additional confirmed by other more sophisticated imaging methods (i.e., MRI), thus I ask the authors provide additional acute effects on cerebral blood flow (CBF) parameters after the first dose treatment. 

The NIRS assessment timeline has been provided in Figure 2.

  1. Also, effects of the dose on hemodynamic responses to brain activation (task performance) on day 1 and day 13 of treatment, that should be provided clearly.

Thank you for the comment. We have added a table in the supplementary material to provide the pre and post data for the hemodynamic responses.

Reviewer 2 Report

The article titled “Incremental Doses of Nitrate-Rich Beetroot Juice Do Not Modify Cognitive Function and Cerebral Blood Flow in Overweight and Obese Older Adults: A 13-Week Pilot Randomised Clinical Trial” by Abrar M Babateen et. al. , discusses the effect of nitrate-rich foods on cognitive function. The article is very interesting and written in fluent and correct English. The authors describe very well the materials and methods used in the study, and provide detailed pictures and graphs to facilitate the understanding of the results. The results support the conclusions of the research. As suggested by the authors, it would be interesting to apply this preliminary study to further studies with more sophisticated imaging methods to obtain even more meaningful data. The authors could implement the introduction, better highlighting the possible role of nutrients and nutraceuticals in counteracting any cognitive pathologies or dysfunctions and could improve the paragraph on both general and age-dependent neurodegeneration. Furthermore, the bibliographic data collected and used, although sufficient, should be updated with more recent articles. For these reasons, I suggest some articles which could be useful in improving the introduction of the manuscript and updating the bibliography.

Dominguez LJ, Barbagallo M. Nutritional prevention of cognitive decline and dementia. Acta Biomed. 2018 Jun 7;89(2):276-290. doi: 10.23750/abm.v89i2.7401. PMID: 29957766; PMCID: PMC6179018.

Fumia A, Cicero N, Gitto M, Nicosia N, Alesci A. Role of nutraceuticals on neurodegenerative diseases: neuroprotective and immunomodulant activity. Nat Prod Res. 2021 Dec 29:1-18. doi: 10.1080/14786419.2021.2020265. Epub ahead of print. PMID: 34963389.

Vlachos GS, Scarmeas N. Dietary interventions in mild cognitive impairment and dementia. Dialogues Clin Neurosci. 2019 Mar;21(1):69-82. doi: 10.31887/DCNS.2019.21.1/nscarmeas. PMID: 31607782; PMCID: PMC6780358.

Carter SJ, Gruber AH, Raglin JS, Baranauskas MN, Coggan AR. Potential health effects of dietary nitrate supplementation in aging and chronic degenerative disease. Med Hypotheses. 2020 Aug;141:109732. doi: 10.1016/j.mehy.2020.109732. Epub 2020 Apr 9. PMID: 32294579; PMCID: PMC7313402.

Domínguez R, Maté-Muñoz JL, Cuenca E, García-Fernández P, Mata-Ordoñez F, Lozano-Estevan MC, Veiga-Herreros P, da Silva SF, Garnacho-Castaño MV. Effects of beetroot juice supplementation on intermittent high-intensity exercise efforts. J Int Soc Sports Nutr. 2018 Jan 5;15:2. doi: 10.1186/s12970-017-0204-9. PMID: 29311764; PMCID: PMC5756374.

Alessio Alesci, Simona Pergolizzi, Patrizia Lo Cascio, Angelo Fumia, Eugenia Rita Lauriano. Neuronal regeneration: Vertebrates comparative overview and new perspectives for neurodegenerative diseases. Acta Zoologica  (IF1.261),  Pub Date : 2021-07-26, DOI: 10.1111/azo.12397

Author Response

Reviewer 2

The article titled “Incremental Doses of Nitrate-Rich Beetroot Juice Do Not Modify Cognitive Function and Cerebral Blood Flow in Overweight and Obese Older Adults: A 13-Week Pilot Randomised Clinical Trial” by Abrar M Babateen et. al. , discusses the effect of nitrate-rich foods on cognitive function. The article is very interesting and written in fluent and correct English. The authors describe very well the materials and methods used in the study, and provide detailed pictures and graphs to facilitate the understanding of the results. The results support the conclusions of the research. As suggested by the authors, it would be interesting to apply this preliminary study to further studies with more sophisticated imaging methods to obtain even more meaningful data. The authors could implement the introduction, better highlighting the possible role of nutrients and nutraceuticals in counteracting any cognitive pathologies or dysfunctions and could improve the paragraph on both general and age-dependent neurodegeneration. Furthermore, the bibliographic data collected and used, although sufficient, should be updated with more recent articles. For these reasons, I suggest some articles which could be useful in improving the introduction of the manuscript and updating the bibliography.

We thank the reviewer for the comment. The introduction has been reviewed and we have added a brief paragraph on the effects of dietary factors on brain health (line 73-81). Thank you for the useful references.

Round 2

Reviewer 1 Report

Dear Authors,
Thanks for revising the manuscript.
I have no additional remarks.